# COPlanner: Plan to Roll Out Conservatively but to Explore Optimistically for Model-Based RL

**Xiyao Wang**[1]    **Ruijie Zheng**[1]    **Yanchao Sun**[2]
**Ruonan Jia**[3]    **Wichayaporn Wongkamjan**[1]    **Huazhe Xu**[3,4]    **Furong Huang**[1]
[1]University of Maryland, College Park    [2]JPMorgan AI Research
[3]Tsinghua University    [4]Shanghai Qi Zhi Institute
{xywang, rzheng12, wwongkam, furongh}@umd.edu
yanchao.sun@jpmchase.com    jiaruonan97@gmail.com
huazhe_xu@mail.tsinghua.edu.cn

## Abstract

Dyna-style model-based reinforcement learning contains two phases: model rollouts to generate sample for policy learning and real environment exploration using current policy for dynamics model learning. However, due to the complex real-world environment, it is inevitable to learn an imperfect dynamics model with model prediction error, which can further mislead policy learning and result in sub-optimal solutions. In this paper, we propose `COPlanner`, a planning-driven framework for model-based methods to address the inaccurately learned dynamics model problem with conservative model rollouts and optimistic environment exploration. `COPlanner` leverages an uncertainty-aware policy-guided model predictive control (UP-MPC) component to plan for multi-step uncertainty estimation. This estimated uncertainty then serves as a penalty during model rollouts and as a bonus during real environment exploration respectively, to choose actions. Consequently, `COPlanner` can avoid model uncertain regions through conservative model rollouts, thereby alleviating the influence of model error. Simultaneously, it explores high-reward model uncertain regions to reduce model error actively through optimistic real environment exploration. `COPlanner` is a plug-and-play framework that can be applied to any dyna-style model-based methods. Experimental results on a series of proprioceptive and visual continuous control tasks demonstrate that both sample efficiency and asymptotic performance of strong model-based methods are significantly improved combined with `COPlanner`. Our code is available at https://github.com/umd-huang-lab/COPlanner.

## 1 Introduction

Model-Based Reinforcement Learning (MBRL) has emerged as a promising approach to improve the sample efficiency of model-free RL methods. Most MBRL methods contain two phases that are alternated during training: 1) the first phase where the agent interacts with the real environment using a policy to obtain samples for dynamics model learning; 2) the second phase where the learned dynamics model rolls out to generate massive samples for updating the policy. Consequently, learning an accurate dynamics model is critical as the model-generated samples with high bias can mislead the policy learning (Deisenroth and Rasmussen, 2011; Wu et al., 2022).

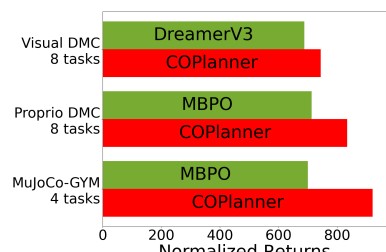

Figure 1: Mean performance of `COPlanner` compared with baselines across 3 diverse benchmarks.

However, dynamics model errors are inevitable due to the complex real-world environment. Existing methods try to avoid model errors in two main ways. 1) Design different mechanisms such as filtering out error-prone samples to mitigate the influence of model errors after model rollouts (Buckman et al., 2018; Yu et al., 2020; Pan et al., 2020; Yao et al., 2021). 2) Actively reduce model errors during

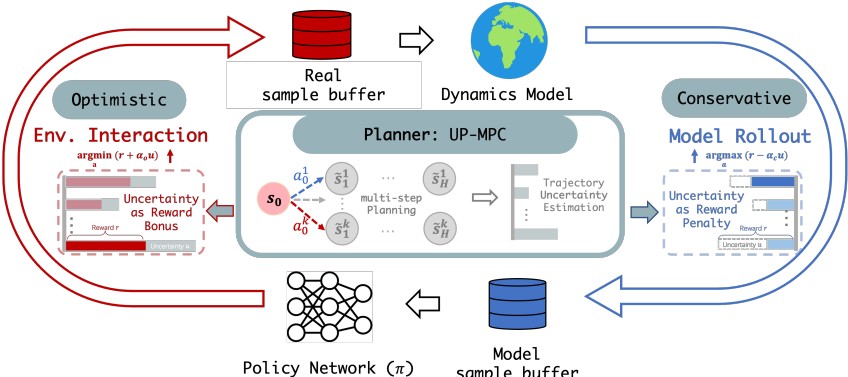

Figure 2: `COPlanner` Framework. The most essential part of `COPlanner` is the *Uncertainty-aware Policy-Guided MPC (UP-MPC) phase* in which we plan trajectories of length $H$, according to the learned dynamics model and learned policy network $\pi$, to select the action with highest trajectory reward. This UP-MPC phase is implemented differently for the two different purposes: *environment exploration* v.s. *dynamics model rollouts*. In environment exploration, trajectory reward has an uncertainty bonus term to encourage exploring uncertain regions in the environment. In dynamics model rollouts, trajectory reward, on the contrary, has an uncertainty penalty term to encourage policy learning on confident regions of the learned dynamics model.

real environment interaction through uncertainty-guided exploration (Shyam et al., 2019; Ratzlaff et al., 2020; Sekar et al., 2020; Mendonca et al., 2021). While both categories of methods have achieved advancements, each comes with its own set of limitations. For the first category, although these approaches are shown to be empirically effective, they primarily concentrate on estimating uncertainty at the current step, often neglecting the long-term implications that present samples might have on model rollouts. Moreover, post-processing samples after model rollouts can compromise rollout efficiency as many model-generated samples are discarded or down-weighted. As for the second category, it is intrinsically challenging to achieve low model error and high long-term reward without sacrificing the sample efficiency by learning exploration policies.

To tackle the aforementioned limitations, we introduce a novel framework, `COPlanner`, which mitigates the model errors from two aspects: 1) avoid being misled by the existing model errors via conservative model rollouts, and 2) keep reducing the model error via optimistic environment exploration. The two aspects are achieved simultaneously by a novel uncertainty-aware multi-step planning method, which requires no extra exploration policy training nor additional samples, resulting in stable policy updates and high sample efficiency. `COPlanner` is structured around three core components: *the Planner*, *conservative model rollouts*, and *optimistic environment exploration*. In the Planner, we employ an *Uncertainty-aware Policy-guided Model Predictive Control* (UP-MPC) to forecast future trajectories in terms of selecting actions and to estimate the long-term uncertainty associated with each action. As shown in Figure 2, this long-term uncertainty serves as dual roles. In the model rollouts phase, the uncertainty acts as a penalty on the total planning trajectory, guiding the selection of conservative actions. Conversely, during the model learning phase, it serves as a bonus on the total planning trajectory, steering towards optimistic actions for environment exploration.

Compared to previous methods, `COPlanner` has the following advantages: **(a)** `COPlanner` has **higher exploration efficiency**, as it focuses on investigating high-reward uncertain regions to broaden the dynamics model, thereby preventing unnecessary excessive exploration of areas with low rewards. **(b)** `COPlanner` has **higher model-generated sample utilization rate.** Through planning for multi-step model uncertainty estimation, `COPlanner` can prevent model rolled out trajectories from falling into uncertain areas, thereby avoiding model errors before model rollouts and improving the utility of model generated samples. **(c)** `COPlanner` enjoys an **unified policy framework**. Unlike previous methods (Sekar et al., 2020; Mendonca et al., 2021) that require training two separate policies for different usage, `COPlanner` only requires training a single policy and we only change the way model-based planning is utilized, thus improving training efficiency and resolving potential policy distribution mismatches. **(d)** `COPlanner` ensures **undistracted policy optimization**. Notably, `COPlanner` diverges from existing approaches by not using long-term uncertainty as an intrinsic reward. Instead, the policy's objective remains focused on maximizing environmental rewards, thereby avoiding the introduction of spurious behaviors due to model uncertainty.

**Summary of Contributions: (1)** We introduce `COPlanner` framework which can mitigate the influence of model errors during model rollouts and explore the environment to actively reduce model errors simultaneously by leveraging our proposed uncertainty-aware policy-guided MPC. **(2)**

`COPlanner` is a plug-and-play framework that can be applicable to any dyna-style MBRL method. **(3)** After being integrated with other MBRL baseline methods, `COPlanner` improves the sample efficiency of these baselines by nearly double. **(4)** Besides, `COPlanner` also significantly improves the performance on a suite of proprioceptive and visual control tasks compared with other MBRL baseline methods (16.9% on proprioceptive DMC, 32.8% on GYM, and 9.6% on visual DMC).

## 2 PRELIMINARIES

**Model-based reinforcement learning.** We consider a Markov Decision Process (MDP) defined by the tuple $(\mathcal{S}, \mathcal{A}, \mathcal{T}, \rho_0, r, \gamma)$, where $\mathcal{S}$ and $\mathcal{A}$ are the state space and action space respectively, $\mathcal{T}(s'|s,a)$ is the transition dynamics, $\rho_0$ is the initial state distribution, $r(s,a)$ is the reward function and $\gamma$ is the discount factor. In model-based RL, the transition dynamics $T$ in the real world is unknown, and we aim to construct a model $\hat{T}(s'|s,a)$ of transition dynamics and use it to find an optimal policy $\pi$ which can maximize the expected sum of discounted rewards,

$$\pi = \operatorname*{argmax}_{\pi} \mathbb{E}_{\substack{s_t \sim \hat{T}(\cdot|s_{t-1},a_{t-1}) \\ a_t \sim \pi(a|s_t)}} \left[\sum_{t=0}^{\infty} \gamma^t r(s_t, a_t)\right]. \tag{1}$$

**Model predictive control.** Model predictive control (MPC) has a long history in robotics and control systems (Garcia et al., 1989; Qin and Badgwell, 2003). MPC find the optimal action through trajectory optimization. Specifically, given the transition dynamics $T$ in the real world, the agent obtains a local solution at each step $t$ by estimating optimal actions over a finite horizon $H$ (i.e., from $t$ to $t + H$) and executing the first action $a_t$ from the computed optimal sequence at time step $t$:

$$a_t = \operatorname*{argmax}_{a_{t:t+H}} \mathbb{E}\left[\sum_{i=t}^{H} \gamma^i r(s_i, a_i)\right], s_i \sim T(\cdot|s_{i-1}, a_{i-1}), \tag{2}$$

where $\gamma$ is typically set to 1. In model-based control methods, the transition dynamics $T$ is simulated by the learned dynamics model $\hat{T}$ (Chua et al., 2018; Wang and Ba, 2019; Hansen et al., 2022).

## 3 THE `COPLANNER` FRAMEWORK

In this section, we will introduce `COPlanner` framework. `COPlanner` consists of three components: the Planner, conservative model rollouts, and optimistic environment exploration. Within the Planner, we propose using an Uncertainty-aware Policy-guided MPC to predict potential future trajectories when selecting different actions under the current state and estimate the long-term uncertainty associated with each action, which will be introduced in Sec 3.1. Depending on the phase, this long-term uncertainty is used to further guide the selection of conservative actions for policy learning or optimistic actions for environment exploration which will be introduced in Sec 3.2 and Sec 3.3.

### 3.1 "THE PLANNER": UNCERTAINTY-AWARE POLICY-GUIDED MPC

In this section, we present the core part of our proposed framework which is called Uncertainty-aware Policy-guided MPC (UP-MPC). Inspired by MPC, we apply the random shooting method (Rao, 2009) to introduce a long-term vision. Specifically, given the current state $s_t$, before each interaction with the

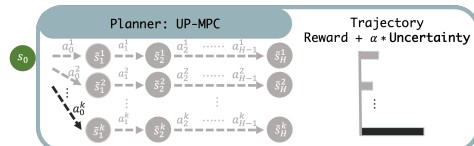

Figure 3: The Planner.

model or real environment, we first generate an action candidate set containing $K$ actions using the policy: $\boldsymbol{a_t} = \{a_t^{(1)}, a_t^{(2)}, ..., a_t^{(k)}\}$. Then, for each action candidate, we perform $H_p$-step planning and calculate the reward $r$, and model uncertainty $u$ for each step. Finally, we select the action according to accumulated reward and model uncertainty, (to interact with the learned dynamics for the model rollouts or to interact with the environment for the model learning), as will be discussed in details in Sec 3.2 and Sec 3.3.

Incorporating model uncertainty is crucial for action selection to compensate for model error. As illustrated in Algorithm 1, we calculate the model uncertainty $u$ through the model disagreement

(Pathak et al., 2019) method. Model disagreement is closely related to model learning and is currently the most common way to estimate model uncertainty in MBRL (Yu et al., 2020; Kidambi et al., 2020; Pan et al., 2020; Sekar et al., 2020; Yao et al., 2021; Mendonca et al., 2021). We train a dynamics model ensemble $\hat{T}_\theta = \{\hat{T}_\theta^{(1)}, \hat{T}_\theta^{(2)}, ..., \hat{T}_\theta^{(n)}\}$ to predict the next state given the current state-action pair $(s_t, a_t)$ as input. Utilizing the ensemble, we approximate the model uncertainty by calculating the variance over predicted states of the different ensemble members. This estimation closely represents the expected information gain (Pathak et al., 2019):

$$u(s_t, a_t) = \frac{1}{N-1} \sum_n (\hat{T}_\theta^{(n)}(s_t, a_t) - \mu')^2, \quad \mu' = \frac{1}{N} \sum_n \hat{T}_\theta^{(n)}(s_t, a_t). \tag{3}$$

See Figure 3 for the illustration of the process. The pseudocode for the Planner, i.e., the UP-MPC process, is summarized in Algorithm 1.

---

**Algorithm 1** The Planner: **UP-MPC** $\left(\pi_\phi, s, \hat{T}_\theta, K, H, \alpha\right)$

---

**Require:** Policy $\pi_\phi$, State $s$, learned dynamics model $\hat{T}_\theta$, number of candidates actions $K$, planning horizon $H_p$, optimistic/conservative parameter $\alpha$
1: Initialize $R^{(k)} = 0$ for $k = 1, ..., K$, $s_0^{(k)} = s$ for $k = 1, ..., K$
2: **for** $k = 1$ to $K$ **do**
3:     **for** $t = 0$ to $H_p - 1$ **do**
4:         Sample $a_t^{(k)} \sim \pi_\phi(\cdot|s_t^{(k)})$
5:         Rollout dynamics model $r_t^{(k)} = \hat{R}(\cdot|s_t^{(k)}, a_t^{(k)}),\ s_{t+1}^{(k)} \sim \hat{T}_\theta(\cdot|s_t^{(k)}, a_t^{(k)})$
6:         Compute model uncertainty $u_t^{(k)}$ according to Eq. 3
7:         $R^{(k)} = R^{(k)} + r_t^{(k)} + \alpha u_t^{(k)}$
8: Select $k^* = \arg\max_{k=1,...,K} R^{(k)}$
9: **return** $a_0^{(k^*)}$

---

Although in Algorithm 1 model uncertainty $u$ is implemented through model disagreement, our proposed UP-MPC is a generic framework, any method for calculating intrinsic rewards to encourage exploration can be embedded into our framework for computing $u$. In Appendix D.5 we provide an ablation study of uncertainty estimation methods to further illustrate this point.

## 3.2 CONSERVATIVE MODEL ROLLOUTS

In model-based RL, due to the limited samples available for model learning, model prediction errors are inevitable. If a policy is trained using model-generated samples with a large error, these samples will not provide correct gradient and may mislead the policy update. Previous methods estimate the model uncertainty of each sample after generation and re-weight or discarded samples with high uncertainty. However, re-weighting samples based on uncertainty still leads to samples with high uncertainty participating in the policy learning process, while filtering requires manually setting an uncertainty threshold, and determining the optimal threshold is difficult. Discarding too many samples can result in inefficient rollouts.

We apply our Planner to plan for maximizing the future reward while minimizing the model uncertainty during model rollouts before executing the action. After calculating the reward and model uncertainty for the $H_p$-step trajectories of $K$ action candidates (line 5 and 6 in Algorithm 1), we replace $\alpha = -\alpha_c$, for a positive $\alpha_c > 0$ at line 7 in Algorithm 1. Mathematically, we select the action according to Eq. 4 to interact with the model for model rollouts:

$$a = \operatorname{argmax}_{a_t \in \boldsymbol{a_t}} \left[ r(s_t, a_t) + \sum_{i=1}^{H_p} r(\hat{s}_{t+i}, \pi(\hat{s}_{t+i})) - \alpha_c \sum_{i=1}^{H_p} u(\hat{s}_{t+i}, \pi(\hat{s}_{t+i})) \right], \hat{s}_{t+i} \sim \hat{T}(\cdot|\hat{s}_{t+i-1}, a_{t+i-1}). \tag{4}$$

The negative $-\alpha_c$ is a coefficient that adds the model uncertainty as a penalty term to the trajectory total reward. By employing this approach, we can prevent model rollout trajectories from falling into model-uncertain regions while obtaining samples with higher rewards.

## 3.3 OPTIMISTIC ENVIRONMENT EXPLORATION

In addition to model rollouts, another crucial part of MBRL is interacting with the real environment to obtain samples to improve the dynamics model. Since the main purpose of MBRL is to improve

sample efficiency, we should acquire more meaningful samples for improving the dynamics model within a limited number of interactions. Therefore, unlike previous methods that merely aimed to thoroughly explore the environment to obtain a comprehensive model (Shyam et al., 2019; Ratzlaff et al., 2020; Sekar et al., 2020), we do not expect the dynamics model to learn all samples in the state space. This is because many low-reward samples do not contribute to policy improvement. Instead, we hope to obtain samples with both high rewards and high model uncertainty to sufficiently expand the model and reduce model uncertainty.

Similar to model rollouts, we also employ our Planner in the process of selecting actions when interacting with the environment. However, the difference lies in that we replace $\alpha = \alpha_o$, for a positive $\alpha_o > 0$ at line 7 in Algorithm 1. Mathematically, we choose the action with both high cumulative rewards and model uncertainty according to Eq. 5, which is a symmetric form of Eq. 4. $\alpha_o$ is a hyperparameter to balance the reward and exploration. Such an action can guide the trajectory towards regions with high rewards and model uncertainty in the real environment, thereby effectively expanding the learned dynamics model.

$$a = \text{argmax}_{a_t \in \boldsymbol{a_t}} \left[ r(s_t, a_t) + \sum_{i=1}^{H_p} r(\hat{s}_{t+i}, \pi(\hat{s}_{t+i})) + \alpha_o \sum_{i=1}^{H_p} u(\hat{s}_{t+i}, \pi(\hat{s}_{t+i})) \right], \hat{s}_{t+i} \sim \hat{T}(\cdot | \hat{s}_{t+i-1}, a_{t+i-1})$$
(5)

In summary, by simultaneously using conservative model rollouts and optimistic environment exploration, `COPlanner` effectively alleviates the model error problem in MBRL. As we will show in Section 5, this is of great help in improving the sample efficiency and performance. The pseudocode of `COPlanner` is shown in Algorithm 2, and a more detailed figure is shown in Appendix A. Very importantly, `COPlanner` achieves both conservative model rollouts and optimistic environment exploration using a single policy. Different from prior exploration methods, the policy that `COPlanner` learns does not have to be an "exploration" policy which is inevitably suboptimal.

---

**Algorithm 2** Main Algorithm: `COPlanner`

---

**Require:** Interaction epochs $I$, rollout horizon $H_r$, planning horizon $H_p$, number of candidates actions $K$, conservative rate $\alpha_c$, optimistic rate $\alpha_o$
1: Initialize policy $\pi_\phi$, dynamics model $\hat{T}$, real sample buffer $\mathcal{D}_e$, model sample buffer $\mathcal{D}_m$
2: **for** $I$ epochs **do**
3:     **while** not Done **do**
4:         Select action $a_t = \textbf{UP-MPC}\big(\pi_\phi, s_t, \hat{T}_\theta, K, H_p, \alpha_o\big)$
5:         Execute in real environment, add $(s_t, a_t, r_t, s_{t+1})$ to $\mathcal{D}_e$
6:         Train dynamics model $\hat{T}_\theta$ with $\mathcal{D}_e$
7:         **for** $M$ model rollouts **do**
8:             Sample initial states from real sample buffer $\mathcal{D}_e$
9:             **for** $h = 0$ to $H_r$ **do**
10:                 Select action $\hat{a}_{t+h} = \textbf{UP-MPC}\big(\pi_\phi, \hat{s}_h, \hat{T}_\theta, K, H_p, -\alpha_c\big)$
11:                 Rollout learned dynamics model and add to $\mathcal{D}_m$
12:         Update current policy $\pi_\phi$ with $\mathcal{D}_m$

---

## 4 RELATED WORK

**Mitigating model error by improving rollout strategies.** Prior methods primarily focus on using dynamics model ensembles (Kurutach et al.; Chua et al., 2018) to assess model uncertainty of samples after they were generated by the model, and then apply weighting techniques (Buckman et al., 2018; Yao et al., 2021), penalties (Kidambi et al., 2020; Yu et al., 2020) or filtering (Pan et al., 2020; Wang et al., 2022) to those high uncertainty samples to mitigate the influence of model error. These methods only quantify uncertainty after generating the samples and since their uncertainty metrics are based on the current step and are myopic, these metrics can not evaluate the potential influence of the current sample on future trajectories. Therefore, they fail to prevent the trajectories, which is generated through model rollout on the current policy, from entering high uncertainty regions, eventually leading to a failed policy update. Wu et al. (Wu et al., 2022) proposed Plan to Predict (P2P), which reverses the roles of the model and policy during model learning to learn an uncertainty-foreseeing model, aiming to avoid model uncertain regions during model rollouts. Combined with MPC, their method achieved promising results. However, their approach lacks effective exploration of the environment. Branched rollout (Janner et al., 2019) and bidirectional rollout (Lai et al., 2020) take advantage of

small model errors in the early stages of rollouts and uses shorter rollout horizons to avoid model errors, but these approaches limit the planning capabilities of the learned dynamics model. Besides, different model learning objectives (Shen et al., 2020; Eysenbach et al., 2022; Wang et al., 2023; Zheng et al., 2023b) are designed to solve objective mismatch (Lambert et al., 2020) in model-based RL and further mitigate model error during model rollouts.

**Reducing model error by improving environment exploration.** Another approach to mitigate model error is to expand the dynamics model by obtaining more diverse samples through exploration during interactions with the environment. However, previous methods mostly focused on pure exploration, i.e., how to make the dynamics model learn more comprehensively (Lowrey et al., 2018; Shyam et al., 2019; Ratzlaff et al., 2020; Sekar et al., 2020; Seyde et al., 2020; Ball et al., 2020; Mendonca et al., 2021; Hu et al., 2023). In complex environments, thoroughly exploring the entire environment is very sample-inefficient and not practical in real-world applications. Moreover, using pure exploration to expand the model may lead to the discovery of many low-reward samples, which are not very useful for policy learning. Mendonca et al. (Mendonca et al., 2021) proposed Latent Explorer Achiever (LEXA) which involves a explorer for exploring the environment and one achiever for solving diverse tasks based on collected samples, but the explorer and achiever may experience policy distribution shift under specific single-task settings, causing the achiever to potentially not converge to the optimal solution.

**Mitigating model error from both sides.** One most relevant work is Model-Ensemble Exploration and Exploitation (MEEE) (Yao et al., 2021) which simultaneously expands the dynamics model and reduces the impact of model error during model rollouts. During the rollout process, it uses uncertainty to weight the loss calculated for each sample to update the policy and the critic. Before interacting with the environment, they first generate $k$ action candidates and then select the action with the highest sum of Q-value and one-step model uncertainty to execute. However, as we mentioned earlier, weighting samples cannot fundamentally prevent the impact of model errors on policy learning, and it may still mislead policy updates. Moreover, since the one-step prediction error of dynamics models is often small (Pan et al., 2020), relying only on the sum of Q-values and one-step model uncertainty may not effectively differentiate action candidates. As a result, samples collected during interactions with the environment might not efficiently expand the model.

## 5 EXPERIMENT

In this section, we conduct experiments to investigate following questions: (a) Can `COPlanner` be applied to both proprioceptive control MBRL and visual control MBRL methods, to improve their sample efficiency and asymptotic performance? (b) How does each component of `COPlanner` impact the performance? (c) How does `COPlanner` influence model learning and model rollouts?

### 5.1 EXPERIMENT ON PROPRIOCEPTIVE CONTROL TASKS

**Baselines**: In this section, we conduct experiments to demonstrate the effectiveness of `COPlanner` on proprioceptive control MBRL methods. We combine `COPlanner` with MBPO (Janner et al., 2019), the most classic method in proprioceptive control dyna-style MBRL, and we name the combined method as **COPlanner-MBPO**. The implementation details can be found in Appendix B. Consequently, **MBPO** naturally becomes one of our baselines. The other three baselines are **P2P-MPC** (Wu et al., 2022), **MEEE** (Yao et al., 2021), and **M2AC** (Pan et al., 2020). These three methods also aim to mitigate the impact of model errors in model-based RL. We choose one of the most popular model-free RL mtheod **D4PG** (Barth-Maron et al., 2018) as another baseline. More details of P2P-MPC, MEEE, and M2AC can be found in Section 4. We also provide comparison with more proprioceptive control MBRL methods in Appendix D.1.

**Environment and hyperparameter settings**: We conduct experiments on 8 proprioceptive continuous control tasks of DeepMind Control (DMC) and 4 proprioceptive control tasks of MuJoCo-GYM (GYM). MBPO trains an ensemble of 7 networks as the dynamics model while using the Soft Actor-Critic (SAC) as the policy network. In COPlanner-MBPO, we adopt the setting of MBPO and directly use the dynamics model ensemble to calculate model uncertainty for action selection in Policy-Guided MPC. For hyperparameter setting, we set optimistic rate $\alpha_o$ to be 1, conservative rate $\alpha_c$ to be 2 in all tasks, and set action candidate number $K$, planning horizon $H_p$ equal to 5 in all tasks. The specific setting are shown in the Appendix C.1.

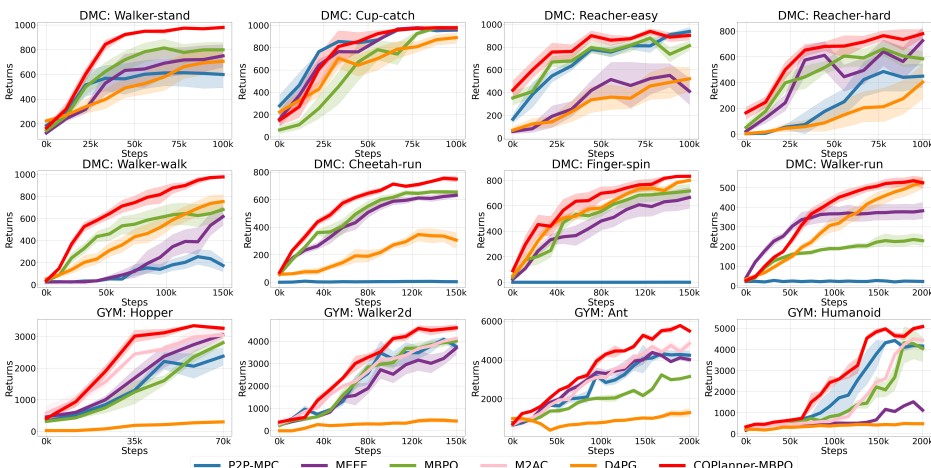

Figure 4: Experiment results of COPlanner-MBPO and other five baselines on proprioceptive control environments. The curves in the first eight figures originate from DM Control tasks, while those in the last four are from GYM tasks. The results are averaged over 8 random seeds, and shaded regions correspond to the 95% confidence interval among seeds. During evaluation, for each seed of each method, we test for up to 1000 steps in the test environment and perform 10 evaluations to obtain an average value. The evaluation interval is every 1000 environment steps.

**COPlanner significantly improves the sample efficiency and performance of MBPO**: Through the results in Figure 4 we can find that both sample efficiency and performance of MBPO have a significant improvement after combining COPlanner. **(a) Sample efficiency**: In proprioceptive control DMC, the sample efficiency is improved by 40% on average compared to MBPO. For example, in the Walker-walk task, MBPO requires 100k steps for the performance to reach 700, while COPlanner-MBPO only needs approximately 60k steps. In more complex GYM tasks, the improvement brought by COPlanner is even more significant. Compared to MBPO, the sample efficiency of COPlanner-MBPO has almost doubled. **(b) Performance**: From the performance perspective, as shown in Figure 1, the performance of MBPO has improved by 16.9% after combining COPlanner. Moreover, it is worth noting that our method successfully solves the Walker-run task, which MBPO fails to address, further demonstrating the effectiveness of our proposed framework. In GYM tasks, the average performance at convergence has increased by 32.8%. Besides, COPlanner-MBPO also outperforms other baselines.

## 5.2 EXPERIMENT ON VISUAL CONTROL TASKS

**Baselines**: We conduct experiments to demonstrate the effectiveness of our proposed framework on visual control environments. We integrate our algorithm with **DreamerV3** (Hafner et al., 2023), the state-of-the-art Dyna-style model-based RL approach recently introduced for visual control. The implementation details can be found in Appendix B. We choose LEXA (Mendonca et al., 2021) as our another baseline. LEXA uses Plan2Explore (Sekar et al., 2020) as intrinsic reward to explore the environment and learn a world model, then using this model to train a policy to solve diverse tasks such as goal achieving. Since pure exploration base on Plan2Explore is sample inefficient for model learning when solving specific tasks, we use the real reward provided by environment as extrinsic reward and add it to intrinsic reward provided by Plan2Explore to train the explorer. We adopt this method on DreamerV3 and call it **LEXA-reward-DreamerV3**. Besides, we compare our method with the SOTA model-free visual RL method **DrQV2** (Yarats et al., 2021). We also provide comparison with more visual control MFRL and MBRL methods in Appendix D.2.

**Environment and hyperparameter settings**: We use 8 visual control tasks of DMC as our environment. In COPlanner-DreamerV3, we learn a latent one-step prediction dynamics model as Plan2Explore (Sekar et al., 2020), the ensemble size is 8. We set action candidate number $K$ and planning horizon $H_p$ equal to 4 in all tasks. For optimistic rate $\alpha_o$ and conservative rate $\alpha_c$, we set them to be 1 and 0.5, respectively. Other hyperparameters remain consistent with DreamerV3 paper.

**COPlanner significantly improves the sample efficiency and performance of DreamerV3**: From the experiment results in Figure 5, we observe that COPlanner-DreamerV3 improves the

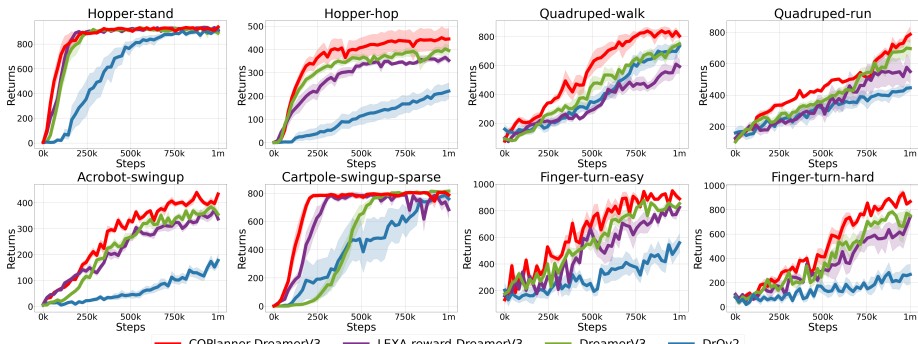

Figure 5: Experiment results of COPlanner-Dreamerv3 and other three baselines on pixel-input DMC. The results are averaged over 8 random seeds, and shaded regions correspond to the 95% confidence interval among seeds. During evaluation, for each seed of each method, we test for up to 1000 steps in the test environment and perform 10 evaluations to obtain an average value. The evaluation interval is every 1000 environment steps.

sample efficiency and performance significantly over DreamerV3, and it demonstrated a significant advantage over DrQv2. The sample efficiency of COPlanner-DreamerV3 is more than twice that of DreamerV3, and the performance is improved by 9.6%. After adding real reward as extrinsic reward for explorer learning, LEXA-reward-DreamerV3 delivers performance comparable to DreamerV3 in most environments. It outperforms DreamerV3 in Cartpole-swingup-sparse and Hopper-stand. However, its performance and sample efficiency are still worse than COPlanner-DreamerV3, further indicates the effectiveness of `COPlanner`.

## 5.3 ABLATION STUDY

In this section, we aim to investigate the impact of different components within `COPlanner` on the sample efficiency and performance. We conduct experiments on two proprioceptive control DMC tasks (Walker-stand and Walker-run) using MBPO as baseline and two visual control DMC tasks (Hopper-hop and Cartpole-swingup-sparse) with DreamerV3 as baseline. The results are demonstrated in Figure 6. Due to page limitations, we provide the ablation study on various hyperparameters of `COPlanner` in Appendix D.4 and the ablation study of uncertainty estimation methods in Appendix D.5.

**From this ablation study, we can see that effectively combining optimistic exploration and conservative rollouts is necessary to achieve the best results.** We find that when only using optimistic exploration (COPlanner w. Explore only), the sample efficiency and performance in all tasks are significantly improved, which highlights the importance of expanding the model. When only using conservative rollouts (COPlanner w. Rollout only), there is some improvement in sample efficiency and performance but to a lesser extent. In more complex visual control tasks, only using conservative rollouts may lead to over-conservatism, resulting in an inability to learn an effective policy in sparse reward environments (as observed with a broken seed in Cartpole-swingup-sparse) or a decrease in sample efficiency during the early stages of learning (Hopper-hop). This is reasonable because conservative rollouts may avoid high uncertainty and high reward areas to ensure the stability of policy updates. Moreover, without efficiently expanding the model, it is challenging to find better solutions using only conservative rollouts in complex visual control tasks. Experimental results show that both optimistic exploration and conservative rollouts are crucial, and using either one individually can lead to an improvement in performance. When combining the two (as in COPlanner), we can achieve the best results, further demonstrating the effectiveness of our method.

## 5.4 MODEL ERROR AND ROLLOUT UNCERTAINTY ANALYSIS

In this section, we will investigate the impact of `COPlanner` on model learning and model rollouts. We provide the curves of how model prediction error and rollout uncertainty change as the environment step increases in Figure 7. We conduct experiments on two proprioceptive control DMC tasks (Cheetah-run and Walker-run) and two visual control DMC tasks (Hopper-hop and Cartpole-swingup-sparse).

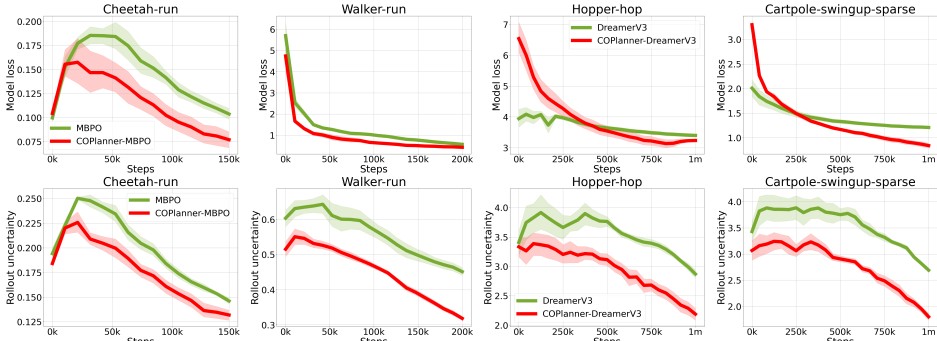

Figure 6: Ablation studies of optimistic exploration and conservative rollouts on different tasks using different mbrl baselines. In the first two proprioceptive control tasks we use MBPO as baseline. For the last two visual control tasks we employ DreamerV3. The results are averaged over 8 random seeds. We can observe that the best results are achieved when combining optimistic exploration and conservative rollouts. The benefit is more pronounced in more-challenging visual tasks.

Figure 7: Model learning loss and rollout uncertainty curves for `COPlanner` and two other model-based RL baselines. The left four are proprioceptive control DMC tasks, and the right four are visual control DMC tasks.

**(1)** In proprioceptive control DMC, we use the MSE loss between model prediction and ground truth next state to evaluate model prediction error during training, while in visual control DMC, we use the KL divergence between the latent dynamics prediction and the next stochastic representation to compute latent model prediction error. We observe that after integrating `COPlanner` in proprioceptive control tasks, the model prediction error is significantly reduced. In more complex visual control tasks, due to obtaining more diverse samples through exploration in the early stages of training, the model prediction error of `COPlanner` is higher than the baseline (DreamerV3). However, as training progresses, the model prediction error rapidly decreases, becoming significantly lower than the model prediction error of DreamerV3. This allows the model to fully learn from the diverse samples, leading to an improvement in policy performance. **(2)** For the evaluation of rollouts uncertainty, we calculate the model disagreement for each sample in the model-generated replay buffer used for policy training using dynamics model ensemble. We find that `COPlanner` significantly reduces rollout uncertainty due to conservative rollouts, suggesting that the impact of model errors on policy learning is minimized. This experiment further demonstrates that the success of `COPlanner` is attributed to both optimistic exploration and conservative rollouts.

## 6 CONCLUSION AND DISCUSSION

We investigate how to effectively address the inaccurate learned dynamics model problem in MBRL. We propose `COPlanner`, a general framework that can be applied to any dyna-style MBRL method. `COPlanner` utilizes Uncertainty-aware Policy-Guided MPC phase to predict the cumulative uncertainty of future steps and symmetrically uses this uncertainty as a penalty or bonus to select actions for conservative model rollouts or optimistic environment exploration. In this way, `COPlanner` can avoid model uncertain areas before model rollouts to minimize the impact of model error, while also exploring high-reward model-uncertain areas in the environment to expand the model and reduce model error. Experiments on a range of continuous control tasks demonstrates the effectiveness of our method. One drawback of `COPlanner` is that MPC can lead to additional computational time and we provide a detailed computational time consumption in Appendix D.6. We can improve computational efficiency by parallelizing planning, which we leave for future work.

ACKNOWLEDGEMENT

Wang, Zheng, and Huang are supported by National Science Foundation NSF-IIS-2147276 FAI, DOD-ONR-Office of Naval Research under award number N00014-22-1-2335, DOD-AFOSR-Air Force Office of Scientific Research under award number FA9550-23-1-0048, DOD-DARPA-Defense Advanced Research Projects Agency Guaranteeing AI Robustness against Deception (GARD) HR00112020007, Adobe, Capital One and JP Morgan faculty fellowships.

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

# Appendix

## A   DETAILED FIGURE OF COPLANNER

We present a more detailed figure to illustrate our COPlanner framework. During environment exploration, we first choose an action using UP-MPC with multi-step uncertainty bonus, then interact with the real environment to obtain real samples for dynamics model learning. In dynamics model rollouts, at each rollout step, we select the actions using UP-MPC with multi-step uncertainty penalty to avoid model uncertain regions and interact with the learned dynamics model to get model-generated samples to update the policy.

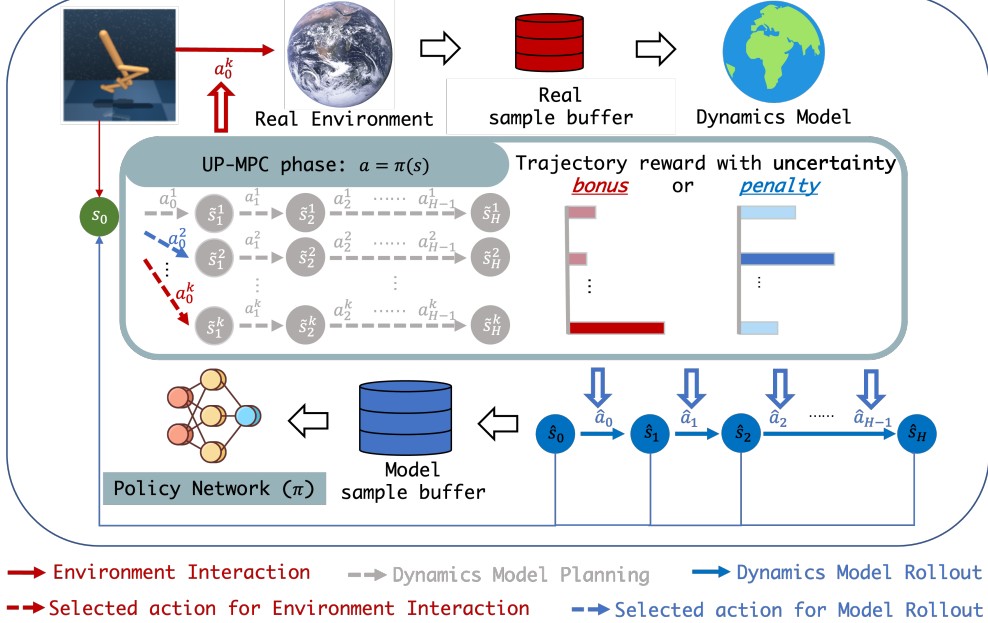

Figure 8: Figure illustration of COPlanner framework with more details.

## B   IMPLEMENTATION

COPlanner framework is versatile and applicable to any dyna-style MBRL algorithm. In this section, we are going to introduce the implementation of two algorithms we used for experiment in Section 5: COPlanner-MBPO for proprioceptive control and COPlanner-DreamerV3 for visual control.

### B.1   COPLANNER-MBPO

MBPO (Janner et al., 2019) trains an ensemble of probabilistic neural networks (Chua et al., 2018) as dynamics model. It utilises negative log-likelihood loss to update each network in the ensemble:

$$\mathcal{L}(\theta) = \sum_{n=1}^{N} [\mu_\theta^b(s_n, a_n) - s_{n+1}]^\top \Sigma_\theta^{b-1}(s_n, a_n)[\mu_\theta^b(s_n, a_n) - s_{n+1}] + \log \det \Sigma_\theta^b(s_n, a_n) \tag{6}$$

For the policy component, MBPO adopts soft actor-critic (Haarnoja et al., 2018). We combine COPlanner with MBPO, the pseudocode is shown in Algorithm 3.

### B.2   COPLANNER-DREAMERV3

DreamerV3 (Hafner et al., 2023) is a dyna-style MBRL method that solves long-horizon tasks from visual inputs purely by latent imagination. Its world model consists of an image encoder, a Recurrent

---

**Algorithm 3** COPlanner-MBPO

---

**Require:** interaction epochs $I$, rollout horizon $H_r$, planning horizon $H_p$, number of candidates actions $K$, conservative rate $\alpha_c$, optimistic rate $\alpha_o$
1: Initialize policy $\pi_\phi$, dynamics model ensemble $\hat{T}_\theta = \{\hat{T}_\theta^1, ..., \hat{T}_\theta^i\}$, real sample buffer $\mathcal{D}_e$, model sample buffer $\mathcal{D}_m$
2: **for** $I$ epochs **do**
3:     **for** t = 1 to T **do**
4:         *// Optimistic environment exploration*
5:         Select action with optimistic rate $a_t = \textbf{UP-MPC}(\pi_\phi, s_t, \hat{T}_\theta, K, H_p, \alpha_o)$
6:         Interact with the real environment with $a_t$, add real sample $(s_t, a_t, r_t, s_{t+1})$ to real sample buffer $\mathcal{D}_e$
7:         Train dynamics model $\hat{T}_\theta$ via Equation 6
8:         **for** $M$ model rollouts **do**
9:             Sample initial rollout states from real sample buffer $\mathcal{D}_e$
10:            **for** h = 0 to $H_r - 1$ **do**
11:                *// Conservative model rollouts*
12:                $\hat{a}_h = \textbf{UP-MPC}(\pi_\phi, \hat{s}_h, \hat{T}_\theta, K, H_p, -\alpha_c)$ (Select action with conservative rate), rollout learned dynamics model and add to model sample buffer $\mathcal{D}_m$
13:         **for** $G$ gradient updates **do**
14:            Update current policy $\pi_\phi$ using model-generated samples from model sample buffer $\mathcal{D}_m$

---

State-Space Model (RSSM) (Hafner et al., 2019b) to learn the dynamics, and predictors for the image, reward, and discount factor. The world model components are:

$$
\begin{aligned}
\text{Recurrent model:} \quad & h_t = f_\phi(h_{t-1}, z_{t-1}, a_{t-1}) \\
\text{Representation model:} \quad & z_t \sim q_\phi(z_t|h_t, x_t) \\
\text{Transition predictor:} \quad & \hat{z}_t \sim p_\phi(\hat{z}_t|h_t) \\
\text{Image predictor:} \quad & \hat{x}_t \sim p_\phi(\hat{x}_t|h_t, z_t) \\
\text{Reward predictor:} \quad & \hat{r}_t \sim p_\phi(\hat{r}_t|h_t, z_t) \\
\text{Discount predictor:} \quad & \hat{\gamma}_t \sim p_\phi(\hat{\gamma}_t|h_t, z_t)
\end{aligned}
$$

where the recurrent model, the representation model, and the transition predictor are components of RSSM. The loss function for the world model learning is:

$$
\begin{aligned}
\mathcal{L}(\phi) = \mathbb{E}_{q_\phi(z_{1:T}|a_{1:T},x_{1:T})}[\sum_{t=1}^{T}(-\ln p_\phi(x_t|h_t, z_t) - \ln p_\phi(r_t|h_t, z_t) - \ln p_\phi(\gamma_t|h_t, z_t) \\
+ \beta_1 max(1, \text{KL}[sg(q_\phi(z_t|h_t, x_t))||p_\phi(z_t|h_t)]) \\
+ \beta_2 max(1, \text{KL}[q_\phi(z_t|h_t, x_t)||sg(p_\phi(z_t|h_t))]))],
\end{aligned}
\tag{7}
$$

where sg means stop gradient. Besides, DreamerV3 also use actor-critic framework as their policy. In particular, they leverage a stochastic actor that chooses actions and a deterministic critic. The actor and critic are trained cooperatively. The actor goal is to output actions leading to states that maximize the critic output, while the critic aims to accurately estimate the sum of future rewards that the actor can achieve from each imagined state (or model rollout state). For more training details about DreamerV3, please refer to their original paper (Hafner et al., 2023).

To estimate model uncertainty in COPlanner-DreamerV3, we train an ensemble of one-step predictive models $\hat{T}_\theta = \{\hat{T}_\theta^1, ..., \hat{T}_\theta^i\}$, each of these models takes a latent stochastic state $z_t$ and action $a_t$ as input and predicts the next latent deterministic recurrent states $h_t$. The ensemble is trained using MSE loss. During model rollouts, we use the world model to generate trajectories, and the one-step model ensemble to evaluate the uncertainty of sample at each rollout step. Here we provide the pseudocode of COPlanner-DreamerV3 in Algorithm 4.

---

**Algorithm 4** COPlanner-DreamerV3

---

**Require:** Rollout horizon $H_r$, planning horizon $H_p$, number of candidates actions $K$, conservative rate $\alpha_c$, optimistic rate $\alpha_o$

1: Initialize real sample buffer $\mathcal{D}_e$ with S random seed episodes.
2: Initialize policy $\pi_\psi$, critic $v_\xi$, one-step model ensemble $\hat{T}_\theta = \{\hat{T}_\theta^1, ..., \hat{T}_\theta^i\}$, world model parameter $\phi$
3: **while** not converged **do**
4:     **for** update step $c = 1..C$ **do**
5:         Draw $\mathcal{B}$ data sequences $\{(a_t, x_t, r_t)\}_{t=k}^{k+L} \sim \mathcal{D}_e$
6:         Compute a latent stochastic states $z_t \sim q_\phi(z_t|h_t, x_t)$
7:         Update world model parameter $\phi$ via Equation 7
8:         *// Conservative model rollouts*
9:         Imagine trajectories $\{(z_\tau, a_\tau)\}_{\tau=t}^{t+H_r}$ from each $z_t$ with $a_\tau = $ **UP-MPC**$\big(\pi_\psi, z_\tau, \hat{T}_\theta, K, H_p, -\alpha_c\big)$.
10:         Update $v_\xi$ and $\pi_\psi$ using imagined trajectories.
11:     **for** time step $t = 1..T$ **do**
12:         Compute $z_t \sim q_\phi(z_t|h_t, x_t)$
13:         *// Optimistic environment exploration*
14:         Select action $a_t = $ **UP-MPC**$\big(\pi_\psi, z_t, \hat{T}_\theta, K, H_p, \alpha_o\big)$.
15:         Interact with the real environment and obatin $(x_t, a_t, r_t, x_t + 1)$
16:     Add experience to $\mathcal{D}_e \leftarrow \mathcal{D}_e \cup \{(x_t, a_t, r_t, x_t + 1)\}_{t=1}^T$

---

## C  HYPERPARAMETERS

In this section, we provide the specific parameters used in each task in our experiments.

Table 1: Hyperparameters of COPlanner-MBPO on proprioceptive control DMC.

| Parameter | Value |
|---|---|
| Conservative rate $\alpha_c$ | 2 |
| Optimistic rate $\alpha_o$ | 1 |
| Action candidate number $K$ | 5 |
| Planning horizon $H_p$ | 5 |
| Real ratio | 0.5 Reacher-xx |
| | 0.8 Finger-spin |
| | 0 Others |
| Rollout horizon $H_r$ | [20, 150, 1, 1] Finger-spin |
| | [20, 150, 1, 4] Others |

### C.1  PROPRIOCEPTIVE CONTROL DMC AND MUJOCO

We use COPlanner-MBPO in all proprioceptive control tasks. For the dynamics model ensemble, we adopted the same setup as MBPO (Janner et al., 2019) original paper, with an ensemble size of 7 and an elite number of 5, which means each time we select the best five out of seven neural networks for model rollouts. Each network in the ensemble is MLP with 4 hidden layers of size 200, using ReLU as the activation function. We train the dynamics model every 250 interaction steps with the environment. The actor and critic structures are both MLP with 4 hidden layers. In proprioceptive control DMC, the hidden layer size of actor and critic is 512, and updated 10 times each environment step, while in MuJoCo the hidden layer size is 256, and they are updated 20 times each environment step. The batch size for model training and policy training is both 256. The learning rate for model training is 1e-3, while the learning rate for policy training is 3e-4.

In MBPO, the authors use samples from both the real sample buffer and the model sample buffer to train the policy, and the ratio of the two is referred to as the real ratio. In addition, MBPO has a unique

mechanism for the rollout horizon $H_r$, which linearly increases with the increase of environment epochs, with each environment epoch including 1000 environment steps. $[a, b, x, y]$ denotes a thresholded linear function, *i.e.* at epoch $e$, rollout horizon is $h = \min(\max(x + \frac{e-a}{b-a}(y-x), x), y)$. The settings for conservative rate $\alpha_c$, optimistic rate $\alpha_o$, action candidate number $K$, planning horizon $H_p$ and the above two parameters in different environments are provided in Table 1 and 2.

Table 2: Hyperparameters of COPlanner-MBPO on MuJoCo.

| Parameter | Value |
|---|---|
| Conservative rate $\alpha_c$ | 2 |
| Optimistic rate $\alpha_o$ | 1 |
| Action candidate number $K$ | 5 |
| Planning horizon $H_p$ | 5 |
| Real ratio | 0.05 |
| Rollout horizon $H_r$ | [20, 100, 1, 4] Hopper |
| | 1 Walker |
| | [20, 150, 1, 15] Ant |
| | [20, 300, 1, 15] Humanoid |

## C.2 VISUAL CONTROL DMC

In Visual control DMC, we use the COPlanner-DreamerV3 method. We keep all parameters consistent with the DreamerV3 original paper (Hafner et al., 2023), except for our newly introduced conservative rate $\alpha_c$, optimistic rate $\alpha_o$, action candidate number $K$, and planning horizon $H_p$. In Table 3, we provide the specific settings of conservative rate $\alpha_c$, optimistic rate $\alpha_o$, action candidate number $K$, and planning horizon $H_p$ for each task. It's worth noting that, although using a conservative rate of 0.5 can perform well, we find that for the two tasks in Quadruped, using a conservative rate of 2 yields the best sample efficiency and performance. For other parameters, please refer to the original DreamerV3 paper. For the one-step predictive model ensemble, we use a model ensemble with ensemble size of 8. Each network in the ensemble is MLP with 5 hidden layers of size 1024.

Table 3: Hyperparameters of COPlanner-DreamerV3 on visual control DMC. We keep all other hyperparameters consistent with the DreamerV3 original paper.

| Parameter | Value |
|---|---|
| Conservative rate $\alpha_c$ | 0.5 |
| Optimistic rate $\alpha_o$ | 1 |
| Action candidate number $K$ | 4 |
| Planning horizon $H_p$ | 4 |

# D MORE EXPERIMENTS

## D.1 COMPARISON WITH MORE PROPRIOCEPTIVE CONTROL MBRL METHODS

In this section, we compared our approach with more proprioceptive control MBRL methods on MuJoCo tasks. In addition to the three baseline methods from Section 5.1, MBPO (Janner et al., 2019), P2P-MPC (Wu et al., 2022), and MEEE (Yao et al., 2021), we introduced two more baselines: PDML (Wang et al., 2023), a method that dynamically adjusts the weights of each sample in the real sample buffer to enhance the prediction accuracy of the learned dynamics model for the current policy, thereby significantly improving the performance of MBPO. And MoPAC (Morgan et al., 2021), a method that also uses policy-guided MPC to reduce model bias. Unlike our approach, MoPAC's policy-guided MPC is solely used for multi-step prediction during rollout based on total reward to

select actions. It does not incorporate a measure of model uncertainty, and therefore, cannot achieve the optimistic exploration and conservative rollouts of COPlanner. The experiment results are shown in Table 4.

As can be seen from Table 4, our method still holds a significant advantage, achieving the best performance in three tasks (Hopper, Walker2d, and Ant). In Humanoid task, it is only surpassed by PDML but is substantially better than the other methods. It's worth mentioning that our approach is orthogonal to PDML, and they can be combined. We believe that by integrating `COPlanner` with PDML, the performance can be further enhanced.

Table 4: Comparison of different MBRL methods on proprioceptive control MuJoCo tasks. Performance is averaged over 8 random seeds.

|  | Hopper (70k) | Walker2d (150k) | Ant (150k) | Humanoid (150k) |
|---|---|---|---|---|
| Ours | **3360.3 ± 165.0** | **4595.3 ± 350.2** | **5268.3 ± 291.4** | 4833.7 ± 320.1 |
| PDML | 3274.2 ± 224.1 | 4378.5 ± 248.9 | 4992.5 ± 365.1 | **5396.7 ± 391.3** |
| MBPO | 2844.6 ± 158.0 | 4221.1 ± 281.1 | 2311.1 ± 252.5 | 1706.0 ± 976.3 |
| P2P-MPC | 2316.8 ± 459.9 | 4151.7 ± 516.9 | 4681.7 ± 591.6 | 3706.1 ± 1360.4 |
| MEEE | 3076.4 ± 165.3 | 3873.8 ± 549.6 | 3932.8 ± 352.7 | 654.2 ± 94.7 |
| MoPAC | 3174.2 ± 233.8 | 2893.6 ± 472.6 | 4382.5 ± 301.7 | 1084.6 ± 573.2 |

Besides, we also conduct comparisons with DreamerV3 and D4PG on six medium-difficulty proprioceptive control DMC tasks, with the experimental results shown in the Figure 9. We find that after integrating with `COPlanner`, both sample efficiency and performance of DreamerV3 are improved significantly.

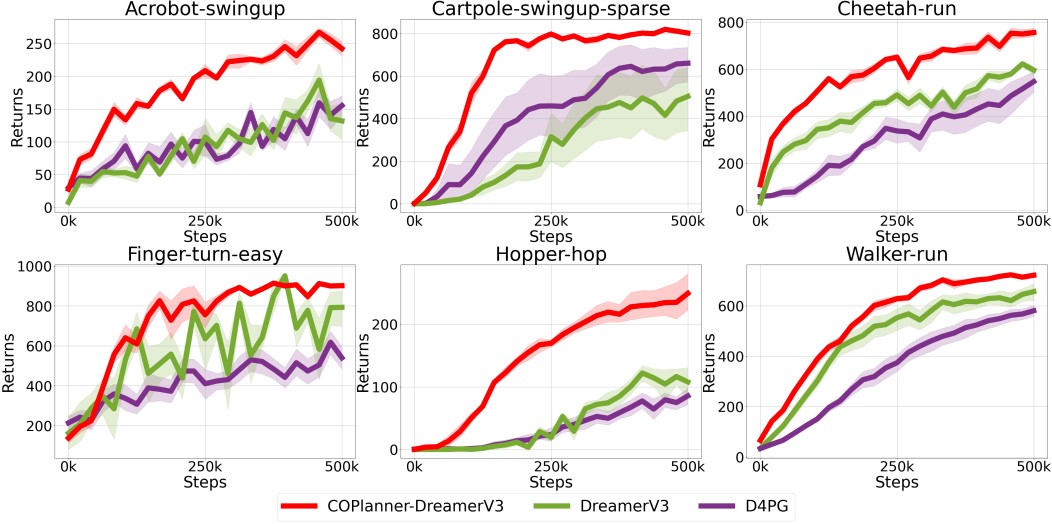

Figure 9: Experiment results of COPlanner-DreamerV3 on 6 medium-difficulty proprioceptive control DMC tasks. The results are averaged over 8 random seeds, and shaded regions correspond to the 95% confidence interval among seeds.

### D.2 COMPARISON WITH MORE VISUAL CONTROL METHODS

In this section, we conducted comparisons with more MFRL and MBRL methods for visual control on 8 tasks from visual DMC. In addition to DreamerV3 (Hafner et al., 2023) and LEXA-reward-DreamerV3 (LEXA-RW), we introduced five more baselines. The first is TDMPC (Hansen et al., 2022). TDMPC learns a task-oriented latent dynamics model and uses this model for planning. During the planning process, TDMPC also learns a policy to sample a small number of actions, thereby accelerating MPC. The second is PlaNet (Hafner et al., 2019b). PlaNet uses the RSSM latent model, which is the same as the Dreamer series (Hafner et al., 2019a; 2020; 2023), and directly uses

this model to perform MPC in the latent space to select actions. Additionally, we also include three more model-free visual RL baselines: TACO (Zheng et al., 2023a), DrM (Xu et al., 2024) and DrQ-v2 (Yarats et al., 2021). The experiment results are shown in Table 5. From the results, it is evident that our method has a significant advantage over all the baselines. In future work, as the representation in `COPlanner` is learned from scratch, we will also integrat pretrained visual representations such as Zheng et al. (2024); Majumdar et al. (2023); Nair et al. (2022) into the `COPlanner` algorithm framework to accelerate the learning process.

Table 5: Performance comparison of different MBRL and MFRL methods on visual DMC tasks at 1 million environment steps.

|  | Hopper-stand | Hopper-hop | Quadruped-walk | Quadruped-run |
|---|---|---|---|---|
| Ours | **937.8 ± 9.7** | **444.2 ± 113.2** | **803.9 ± 59.3** | **787.9 ± 111.9** |
| DreamerV3 | 888.5 ± 60.7 | 395.3 ± 50.1 | 749.2 ± 65.3 | 696.0 ± 132.2 |
| LEXA-RW | 905.7 ± 25.1 | 353.0 ± 33.8 | 591.0 ± 110.9 | 549.5 ± 186.2 |
| TDMPC | 821.6 ±70.8 | 189.2 ± 19.7 | 427.8 ± 50.2 | 393.8 ± 40.9 |
| PlaNet (5m) | 5.96 | 0.37 | 238.90 | 280.45 |
| TACO | 923 ± 22 | 261 ± 52 | 798 ± 15 | 531 ± 38 |
| DrM | 910 ± 15 | 162 ± 82 | 750 ± 29 | 452 ± 32 |
| DrQ-v2 | 918 ± 12 | 192 ± 41 | 702 ± 42 | 407 ± 21 |
|  | Acrobot-swingup | Cartpole-swingup-sparse | Finger-turn-easy | Finger-turn-hard |
| Ours | **434.4 ± 31.0** | 787.5 ± 37.5 | **889.8 ± 28.3** | **867.1 ± 50.6** |
| DreamerV3 | 354.9 ± 37.9 | **814.7 ± 22.3** | 853.9 ± 37.5 | 751.3 ± 93.1 |
| LEXA-RW | 337.7 ± 24.9 | 683.5 ± 145.6 | 825.5 ± 55.9 | 740.7 ± 162.0 |
| TDMPC | 227.5 ± 16.9 | 668.3 ± 49.1 | 703.8 ± 65.2 | 402.7 ± 112.6 |
| PlaNet (5m) | 3.21 | 0.64 | 451.22 | 312.55 |
| TACO | 241 ± 21 | 802 ± 21 | 752 ± 45 | 618 ± 17 |
| DrM | 228 ± 15 | 799 ± 7 | 692 ± 39 | 459 ± 36 |
| DrQ-v2 | 210 ± 12 | 794 ± 14 | 576 ± 42 | 248 ± 52 |

### D.3 EXPERIMENTS COMBINED WITH DREAMERV2

We also combine `COPlanner` with DreamerV2 (Hafner et al., 2020) for experimentation, with the results shown in Figure 10. After integrating with DreamerV2, our method also achieves a significant improvement in both sample efficiency and performance.

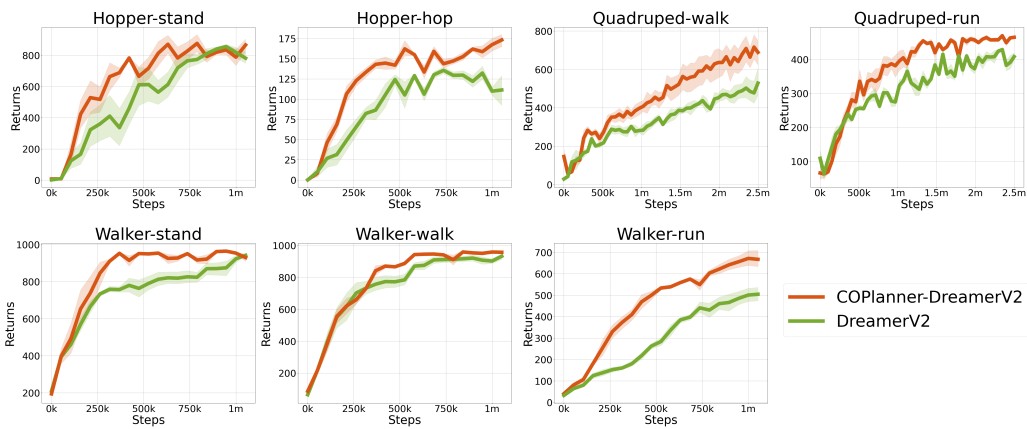

Figure 10: Experiment results of COPlanner-DreamerV2 on 7 visual control DMC tasks. The results are averaged over 4 random seeds, and shaded regions correspond to the 95% confidence interval among seeds. During evaluation, for each seed of each method, we test for up to 1000 steps in the test environment and perform 10 evaluations to obtain an average value. The evaluation interval is every 1000 environment steps.

### D.4 Hyperparameter study

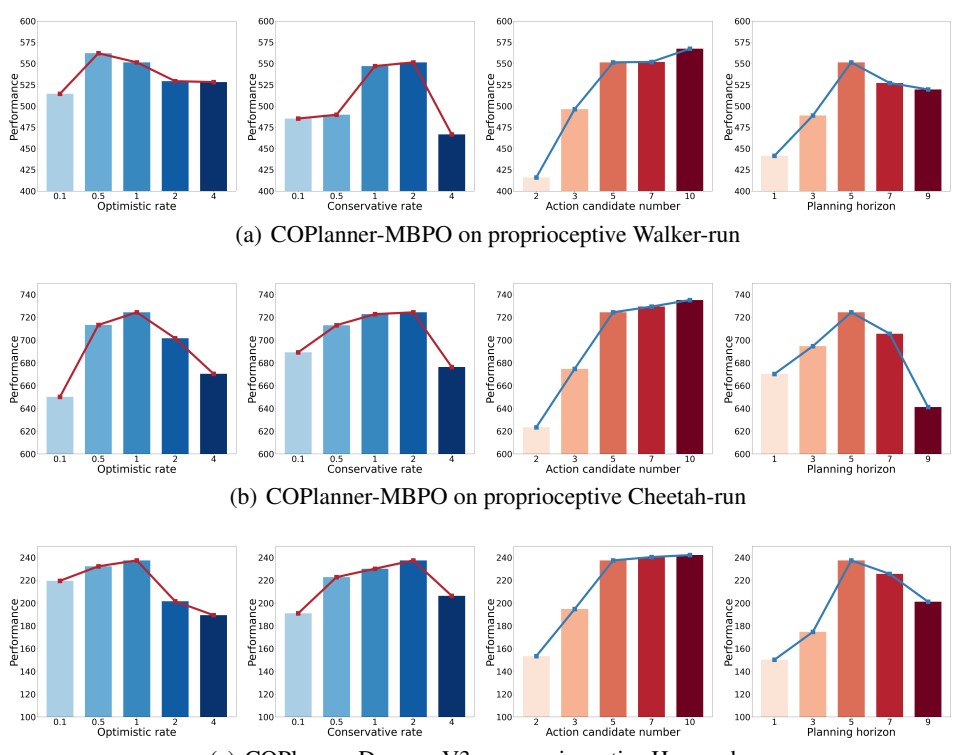

(a) COPlanner-MBPO on proprioceptive Walker-run

(b) COPlanner-MBPO on proprioceptive Cheetah-run

(c) COPlanner-DreamerV3 on proprioceptive Hopper-hop

Figure 11: Ablation studies of COPlanner's different hyperparameters. Experiments are conducted using COPlanner-MBPO on proprioceptive Walker-run and proprioceptive Cheetah-run, and using COPlanner-DreamerV3 on proprioceptive Hopper-hop. The results are averaged over 4 random seeds. From left to right, the results are for different parameters of optimistic rate $\alpha_o$, conservative rate $\alpha_c$, action candidate number $K$, and planning horizon $H_p$.

In this section, we conduct hyperparameter studies to investigate the impact of different hyperparameters on COPlanner. We perform experiments on the Walker-run and Cheetah-run task of proprioceptive control DMC using COPlanner-MBPO, and on the Hopper-hop using COPlanner-DreamerV3. The original hyperparameter settings are: optimistic rate $\alpha_o$ is 1, conservative rate $\alpha_c$ is 2, action candidate number $K$ is 5, and planning horizon $H_p$ is 5. When conducting ablation experiments for each hyperparameter, other parameters remain unchanged. The results are shown together in Figure 11.

**Optimistic rate $\alpha_o$:** we observe that the best $\alpha_o$ lies between 0.5 to 1. When the $\alpha_o$ is too large, COPlanner tends to excessively explore high uncertainty areas while neglecting rewards, leading to a decrease in sample efficiency and performance. On the other hand, when the $\alpha_o$ is too small, COPlanner fails to achieve the desired exploration effect.

**Conservative rate $\alpha_c$:** the optimal range for the $\alpha_c$ is between 1 and 2. A too large $\alpha_c$ may lead to overly conservative selection of low-reward actions, while a too small $\alpha_c$ would be unable to make model rollouts avoid model uncertain areas.

**Action candidate number $K$:** we find that $K$ has a significant impact on sample efficiency and performance. When $K$ is set to 2, the improvement of COPlanner over MBPO in terms of performance and sample efficiency is relatively limited. This is reasonable because if there are only a few action candidates, our selection space is very limited, and even with the use of uncertainty bonus and penalty to select actions, there may not be much difference. When $K$ increases to more than 5, the effect of COPlanner becomes very stable, and more candidates do not bring noticeable improvements in performance and sample efficiency.

**Planning horizon $H_p$:** when $H_p$ is 1, we find that `COPlanner`'s improvement on performance and sample efficiency is relatively limited. This also confirms what we mentioned in Section 1: only considering the current step while ignoring the long-term uncertainty impact cannot completely avoid model errors, as samples with low current model uncertainty might still lead to future rollout trajectories falling into model uncertain regions. As the planning horizon gradually increases, performance and sample efficiency also rise. When the planning horizon is too long ($H_p$ equals to 7 or 9), it is possible that due to the bottleneck of the model planning capability, most action candidates' corresponding trajectories fall into model uncertain areas, leading to a slight decline in performance and sample efficiency.

### D.5 ABLATION STUDY OF MODEL UNCERTAINTY ESTIMATION METHODS

We conduct an ablation study on the Hopper-hop task in visual control DMC to evaluate different uncertainty estimation methods. We adopt two methods, RE3 (Seo et al., 2021) and MADE (Zhang et al., 2021), which are used to estimate intrinsic rewards in pixel input, to replace the disagreement in calculating $u(s, a)$ in Equation 4 and 5. The results are shown in Figure 12. We find that the performance achieved using these two methods is similar to that of disagreement. This demonstrate that using disagreement to calculate uncertainty is not the primary reason for the observed performance improvement.

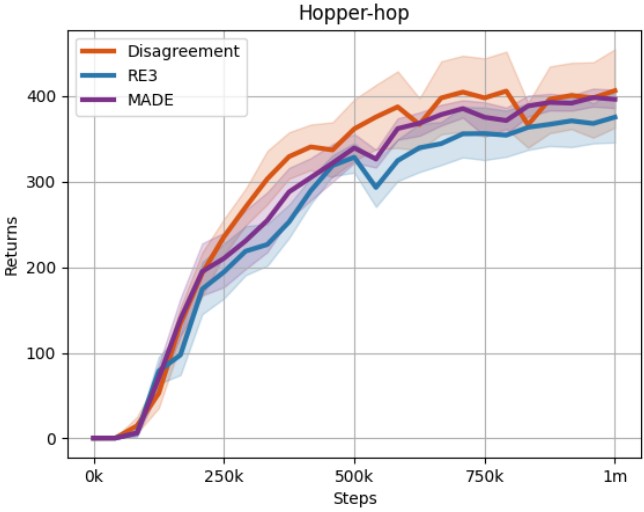

Figure 12: Ablation study of different uncertainty estimation methods.

### D.6 COMPUTATIONAL TIME CONSUMPTION OF `COPLANNER`

We provide a comparison of the computational time consumption between the baseline methods and `COPlanner` across different domains in Table 6. All timings are reported using a single NVIDIA 2080ti GPU.

Table 6: Average time consumption (h).

|  | MuJoCo | Proprioceptive DMC | Visual DMC |
|---|---|---|---|
| COPlanner-MBPO | 41.2 | 11.3 | N/A |
| MBPO | 33.78 | 10.6 | N/A |
| COPlanner-DreamerV3 | N/A | N/A | 17.9 |
| DreamerV3 | N/A | N/A | 13.1 |

### D.7 DIVERSITY EVALUATION OF REAL SAMPLE BUFFER

In this section, to further demonstrate that our method achieves better exploration of the environment, we evaluate the average state entropy of the real sample buffer obtained using our method and DreamerV3 at 1 million environment steps. A higher average state entropy implies that the real sample buffer covers more states in the real environment, indicating that the samples in the real sample buffer are more diverse and thus suggesting more thorough exploration of the environment (Seo et al., 2021). We conduct evaluations on four visual DMC tasks include Hopper-hop, Quadruped-walk, Acrobot-swingup, and Finger-turn-hard. Following the work of RE3 (Seo et al., 2021), in order to estimate state entropy in environments with high-dimensional observations, we utilize a k-nearest neighbor entropy estimator in the low-dimensional representation space of a randomly initialized encoder. Our encoder consists of three convolutional layers with 3x3 kernels, a stride of 2, and padding of 1, followed by a flattening layer. The activation function between each layer is ReLU. After passing through the encoder, each image from the replay buffer is compressed into a 512-dimensional latent state, and the k-nearest neighbor state entropy is estimated as follows:

$$e(s_i) = log(||y_i - y_i^{k-NN}||_2 + 1), \tag{8}$$

where where $y_i = f_\theta(s_i)$ is a fixed representation from a random encoder and $y_i^{k-NN}$ is the k-nearest neighbor of $y_i$ within a set of $N$ representations $\{y_1, y_2, ..., y_n\}$. We set $N = 1024$. Then we average the k-nearest neighbor state entropy of each sample in real sample buffer. The final results are shown in Figure 13.

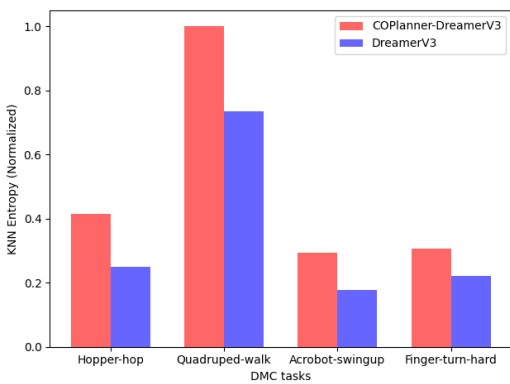

Figure 13: K-nearest neighbor state entropy estimation on different visual DMC tasks compared with DreamerV3.

The figure clearly shows that the state entropy of the real sample buffer significantly increases after integrating our method. This indicates that the real samples obtained by our method are more diverse, achieving better exploration of the environment.

### D.8 VISUALIZATION OF EXPERIMENT RESULTS

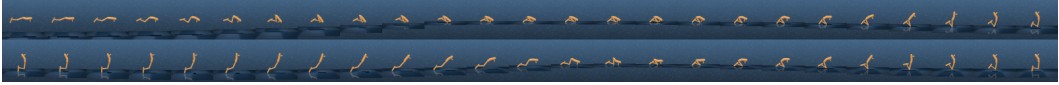

Figure 14: Visualization of policy learned by DreamerV3 on Hopper-hop.

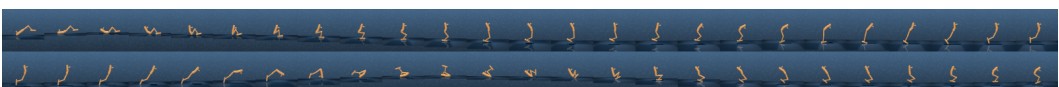

Figure 15: Visualization of policy learned by COPlanner-DreamerV3 on Hopper-hop.

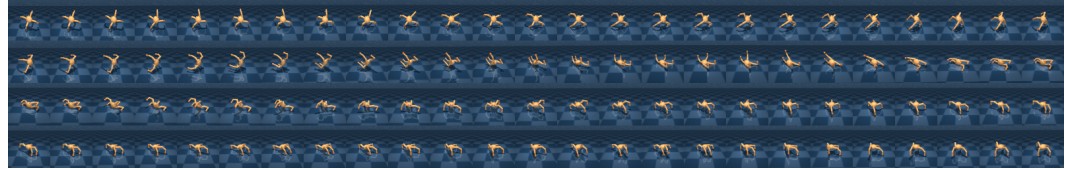

Figure 16: Visualization of policy learned by DreamerV3 on Quadruped-walk.

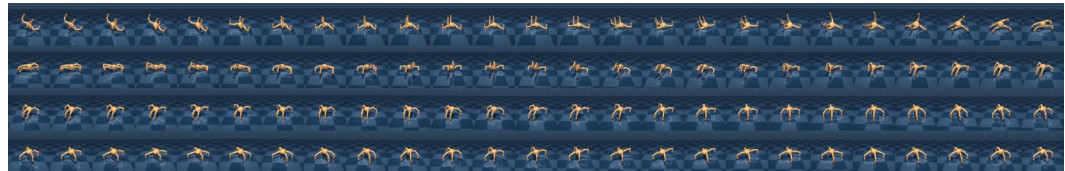

Figure 17: Visualization of policy learned by COPlanner-DreamerV3 on Quadruped-walk.

In this section, to better demonstrate the improvements brought by our method, we visualize the trajectories obtained from evaluations in the real environment after convergence using DreamerV3 and our method (COPlanner-DreamerV3). We visualize trajectories in Hopper-hop and Quadruped-walk tasks, as shown in Figure 14, 15, 16, and 17.

From the comparison between Figure 14 and Figure 15, we can observe that in the Hopper hop task, DreamerV3 is only able to learn to jump using the knee, whereas our method can learn to jump using the feet and perform somersaults during the jump. Through the comparison of Figure 16 and Figure 17, we can see that in the Quadruped-walk task, our method is able to learn a more stable behavior of walking using all four legs, as opposed to DreamerV3, which learns to walk using only three legs.

