# OpenReview forum: "COPlanner: Plan to Roll Out Conservatively but to Explore Optimistically for Model-Based RL"
_ICLR.cc/2024/Conference — ICLR 2024 poster_

### Official Review · Reviewer_D4cs · 2023-10-23

**Soundness:** 3 good
**Presentation:** 3 good
**Contribution:** 2 fair
**Rating:** 6
**Confidence:** 4

**Summary:**

This work presents COPlanner, a model-based reinforcement learning algorithm that combines optimistic exploration of high-reward model uncertain regions with pessimistic model learning to avoid biased updates in information-sparse regions. The approach leverages an MPC-style action selection and is evaluated on both proprioceptive as well as visual control tasks.

**Strengths:**

-	Increasing learning efficiency of model-based reinforcement learning agents is an important research direction, particularly in light of hardware deployment under costly data generation
-	Combining optimistic exploration of uncertain high-reward behaviors with conservative model rollouts to improve information quality of samples is a very promising research direction
-	MPC-style action selection provides several benefits over direct policy evaluation
- The approach is evaluated against several baseline agents on OpenAI Gym as well as DeepMind Control Suite environments, with both proprioceptive and visual tasks for the latter

**Weaknesses:**

-	Generally, the paper would benefit from a stronger selection of established baselines. E.g., for proprioceptive DMC model-free MPO/DMPO/D4PG or model-based DreamerV3, for visual DMC DrQ-v2 as a model-free baseline
-	The visual control experiments should be run for more than 1M steps as agents have not converged on the Acrobot/Finger/Quadruped tasks. Comparing with Dreamer-v1 Figure 10, agents would be expected to converge and solve the tasks at approximately 2M steps.
-	The Dreamer-v2 scores in Figure 9 appear to be significantly lower than the ones provided by the original authors on GitHub – which Dreamer implementation was used to obtain these scores?
-	The task selection for proprioceptive DMC consists of relatively easy tasks and some of the “MuJoCo”/Gym tasks have not been run to convergence
-	The hyperparameter study in Appendix D.4 is interesting, but should be extended to more environments to observe clearer trends
-	Some highly related work is missing. E.g., the optimistic exploration of uncertain future returns for visual control has previously been investigated in [1], also building on the Dreamer-v2 agent, while optimistic finite-horizon exploration under nominal reward functions was studied in [2]. Additionally, work such as POLO [3] would be relevant.

[1] T. Seyde, et al. "Learning to plan optimistically: Uncertainty-guided deep exploration via latent model ensembles," CoRL 2021.

[2] P. Ball, et al. "Ready policy one: World building through active learning," ICML 2020.

[3] K Lowrey, et al.  "Plan online, learn offline: Efficient learning and exploration via model-based control," ICLR 2019.


Minor:

-	The DMC tasks are MuJoCo-based, so “MuJoCo” task description should be replaced by “Gym”

**Questions:**

-	Action selection by optimizing over a 5-step rollout without terminal value function is surprising. Do you have intuition for why this short-term reasoning is sufficient?
-	The original Dreamer-v3 paper also evaluated on proprioceptive DMC environments - why not use Dreamer-v3 as a proprioceptive baseline as well?
-	There are several task-specific parameter choices for proprioceptive control. How impactful are these? Ideally a single set of hyperparameters would be used throughout.

---

> ### Author Response · Authors · 2023-11-17
> **Response to Reviewer D4cs Part 1**
>
> We thank Reviewer D4cs for the insightful feedback. We are encouraged that Reviewer D4cs finds our idea is important and promising, and experiments are fruitful. We now address all of Reviewer D4cs's concerns and questions below:
>
> > Weakness 1: Generally, the paper would benefit from a stronger selection of established baselines. E.g., for proprioceptive DMC model-free MPO/DMPO/D4PG or model-based DreamerV3, for visual DMC DrQ-v2 as a model-free baseline.
>
> Thank you for your suggestion. Following your advice, we add a comparison with D4PG in the proprioceptive control task in Figure 4, and a comparison with DrQv2 in the visual control task in Figure 5.
>
> Additionally, due to space limitations, we include a comparison with DreamerV3 in six medium-difficulty proprioceptive control tasks in Appendix D.1. For specific experimental details and results, please refer to General Response and the updated paper.
>
> > Weakness 2: The visual control experiments should be run for more than 1M steps as agents have not converged on the Acrobot/Finger/Quadruped tasks. Comparing with Dreamer-v1 Figure 10, agents would be expected to converge and solve the tasks at approximately 2M steps.
>
> We would like to offer our apologies here. Due to differences in the counting methods for Counter and Environment Step in the JAX code of DreamerV3, there is a mistake in the previous version of our paper. The original version of Figure 5 represents the performance of all methods at 500k environment steps. We have run these methods up to 1M steps and have reported the new results in both Figure 5 and Table 5 in the revised manuscript. The new results show that nearly all methods have converged at 1M steps, and DreamerV3 significantly outperforms the results reported in the DreamerV1 paper. We apologize again for this mistake.
>
> > Weakness 3: The Dreamer-v2 scores in Figure 9 appear to be significantly lower than the ones provided by the original authors on GitHub – which Dreamer implementation was used to obtain these scores?
>
> We use one of the most popular PyTorch implementations of DreamerV2 available on GitHub, which has the second-highest number of stars: https://github.com/jsikyoon/dreamer-torch. Since it is implemented in PyTorch, there may be differences in the results compared to the TensorFlow version of the original source code. Therefore, we only include these experiments in the Appendix as a reference and do not feature them in the main paper. In the main paper, for the DreamerV3 experiments, we use the original JAX code released with the DreamerV3 paper.
>
> > Weakness 4: The task selection for proprioceptive DMC consists of relatively easy tasks and some of the “MuJoCo”/Gym tasks have not been run to convergence
>
> The reason we chose these tasks from proprioceptive DMC is that, based on our extensive testing, MBPO-style methods do not work well in more complex DMC tasks. Therefore, we only report the tasks where methods based on MBPO are effective.
>
> As an addition, in the updated version of our paper, we include a comparison with DreamerV3 in six medium-difficulty proprioceptive control DMCtasks to demonstrate that our method can improve the performance and sample efficiency over the baseline in more complex tasks. Furthermore, in the updated paper, we run the Ant and Humanoid environments from OpenAI Gym up to 200k steps, where all methods have now converged. For specific experimental results, please refer to the updated paper.
>
> > Weakness 5: The hyperparameter study in Appendix D.4 is interesting, but should be extended to more environments to observe clearer trends
>
> In the updated version of our paper, we add two sets of hyperparameter studies, one using COPlanner-MBPO on the Cheetah-run task and the other using COPlanner-DreamerV3 on the Hopper-hop task. The trends we observed are largely consistent with our previous experimental results. For specific experimental results, please refer to **Appendix D.4, Figure 11** in the updated paper.
>
> > Weakness 6: Some highly related work is missing.
>
> Thank you for your reminder. We have added references to these papers in the related works section of our updated paper.

---

> > ### Author Response · Authors · 2023-11-17
> > **Response to Reviewer D4cs Part 2**
> >
> > > Question 1: Action selection by optimizing over a 5-step rollout without terminal value function is surprising. Do you have intuition for why this short-term reasoning is sufficient?
> >
> > This is indeed an interesting point we've noted as well. We experimented with incorporating a terminal value function into the MPC process, but found that it did not offer significant improvements over our current method (for both proprioceptive and visual control). Additionally, since we perform planning to estimate long-term uncertainty, introducing a terminal value function would also require estimating the uncertainty of the terminal value, which would further increase computational costs. For these reasons, we did not include a terminal value function.
> >
> > As for why incorporating a terminal value function didn't work, we suspect it's because model compounding errors lead to significant inaccuracies in the state predicted by the model after several rollout steps. Consequently, the value function may not provide an accurate value estimation. Furthermore, the scale of state value is much larger than the rewards and estimated uncertainty, which might result in the state value solely dictating the MPC's action choices, rendering our method ineffective. Therefore, we believe short-term reasoning is a better solution. In future work, we plan to further explore how to integrate long-term reasoning into our framework.
> >
> > > Question 2: The original Dreamer-v3 paper also evaluated on proprioceptive DMC environments - why not use Dreamer-v3 as a proprioceptive baseline as well?
> >
> > We choose to use MBPO because we want to demonstrate that our method can be combined with any dyna-style MBRL method, not just Dreamer. MBPO is one of the most representative MBRL methods in proprioceptive control. As a supplement, we add a comparison with DreamerV3 in proprioceptive control tasks in **Appendix D.1**. The experimental results show that our method still significantly improves both performance and sample efficiency compared to DreamerV3 in proprioceptive control tasks.
> >
> > > Question 3: There are several task-specific parameter choices for proprioceptive control. How impactful are these? Ideally a single set of hyperparameters would be used throughout.
> >
> > Due to the time constraints of the submission and the longer experimental duration required for MBPO-based methods, we did not unify the parameters for GYM in the initial paper. However, **in the updated paper, we have unified the experimental hyperparameters, and now for all proprioceptive control experiments, the hyperparameters are the same (optimistic rate=1, conservative rate=2, action candidate number=5, planning horizon=5)**. Please refer to Figure 4 for the updated experimental results. We find that after unifying the hyperparameters, the performance of COPlanner in GYM's Hopper, Walker, and Ant environments has improved compared to the previous version. The performance in Humanoid remains largely unchanged. This is consistent with the conclusions of our hyperparameter studies, as for Humanoid, the only change was in the conservative rate from 1 to 2, and this parameter's setting between 1 and 2 has little difference in performance across different environments.
> >
> > -----
> > Thank you again for your effort in reviewing our paper! We are happy to answer any further questions.

---

> ### Author Response · Authors · 2023-11-20
> **Does our response address your concerns?**
>
> Dear reviewer D4cs,
>
> As the stage of the review discussion is ending soon, we would like to kindly ask you to review our revised paper as well as our response and consider making adjustments to the scores. We believe we have addressed all concerns. Please let us know if there are any other questions. We would appreciate the opportunity to engage further if needed.

---

> > ### Comment · Reviewer_D4cs · 2023-11-20
> > **Response to Rebuttal**
> >
> > Thank you for your replies and updated experiments. I still believe that extension to more complex tasks would benefit the potential impact of the paper. Regarding integration of additional related work: RP1 (Ball et al.) and LOVE (Seyde et al.) explore uncertain rewards and not purely uncertain dynamics. There are some important differences between the works listed on page 6 [Lowrey, ..., Hu] which would be worth expanding on as well as relating to the presented work. Overall, the updated experiments look significantly stronger and I'm leaning towards accept.

---

### Official Review · Reviewer_NmVv · 2023-11-01

**Soundness:** 3 good
**Presentation:** 3 good
**Contribution:** 2 fair
**Rating:** 6
**Confidence:** 3

**Summary:**

This paper proposes a model-based reinforcement learning framework COPlanner to mitigate the model errors from the model rollouts and environment exploration. This framework includes three components, and the most crucial component is the Planner to predict future trajectories based on selected actions and their corresponding uncertainties, and the uncertainties work as penalties during model rollouts and bonuses during environment exploration.

**Strengths:**

1. The paper is easy to understand.
2. The idea of using the variance over predicted states of the ensemble members to approximate the model uncertainty either penalty or reward seems interesting to me.
3. The proposed framework outperforms baselines in almost all experiments.

**Weaknesses:**

1. Lack of comparisons to some framework.  In section 3.2, the paper mentions some previous methods to estimate uncertainty samples after generations or to decrease model error by re-weighting or discarding samples with high uncertainty. This should require a comparison to demonstrate using variance through model ensemble.
2. Lack of visualization of experiments. The results are all basically tables, line plots.

**Questions:**

1. It's still not quite clearly to me how Conservative rate and Optimistic rate are selected. What's the intuition here about why they are always different than each other in every experiment settings? Are they selected according to Action candidate number, and Planning horizon as well?
2. Following the weakness, experiments for comparison and more visualizations are necessary.

---

> ### Author Response · Authors · 2023-11-17
> **Response to Reviewer NmVv**
>
> We thank Reviewer NmVv for the insightful feedback. We are encouraged that Reviewer NmVv finds our paper easy to follow, the idea interesting, and experiment results strong. We now address all of Reviewer NmVv's concerns and questions below:
>
> > Weakness 1: Lack of comparisons to some framework. In section 3.2, the paper mentions some previous methods to estimate uncertainty samples after generations or to decrease model error by re-weighting or discarding samples with high uncertainty. This should require a comparison to demonstrate using variance through model ensemble.
>
> In the updated version of our paper, we include experimental comparisons with M2AC [1]. M2AC falls under the category of methods that discard samples with high uncertainty. For the category of methods that re-weight samples with high uncertainty, one of our comparison baselines, MEEE, belongs to this category. The additional experimental results also demonstrate the strong performance of our method. For specific experimental details and results, please refer to Section 5.1 and Figure 4 in the updated version of the paper.
>
>
> > Weakness 2: Lack of visualization of experiments. The results are all basically tables, line plots.
>
>
> In the updated version of our paper, we include experiments on test time trajectory visualization. We visualize the trajectories of policies learned by COPlanner-DreamerV3 and Dreamer V3 in the test environment after convergence of policy learning in **Appendix D.8**. From Figures 14 to 17, it can be seen that COPlanner-DreamerV3 learned more rational and stable behaviors compared to DreamerV3.
>
> > Question 1: It's still not quite clearly to me how Conservative rate and Optimistic rate are selected. What's the intuition here about why they are always different than each other in every experiment settings? Are they selected according to Action candidate number, and Planning horizon as well?
>
> As discussed in Appendix D.4 of our paper, we conduct a detailed hyperparameter study to discuss how parameters should be chosen and then selected a set of parameters that perform relatively well across all tasks. Regarding why these two parameters are different, we believe that these two are essentially completely unrelated parameters, one controlling exploration and the other controlling exploitation (reducing model error). In proprioceptive control tasks, since the model directly predicts the next state, it is more prone to model compounding errors, thus requiring a larger Conservative rate. In contrast, in visual control tasks, the dynamics model predicts in the latent space, which to some extent mitigates the impact of model compounding errors. Therefore, the Conservative rate can be set relatively lower, allowing the model rollout to more likely choose high-reward, model-generated samples for training the policy. The choice of these two parameters is independent of Action candidate number and Planning horizon.
>
> Reference:
>
> [1]. "Trust the Model When It Is Confident: Masked Model-based Actor-Critic." Pan et. al. NeurIPS 2020.
>
>
> -----
> Thank you again for your effort in reviewing our paper! We are happy to answer any further questions.

---

> > ### Comment · Reviewer_NmVv · 2023-11-21
> >
> > Thanks for the clarifications! I choose to maintain my current score.

---

### Official Review · Reviewer_EHTZ · 2023-11-01

**Soundness:** 3 good
**Presentation:** 3 good
**Contribution:** 2 fair
**Rating:** 6
**Confidence:** 3

**Summary:**

This paper proposes a model based reinforcement learning method that target for solving the challenge of inaccurately learned dynamic model problem through a combination of conservative model rollouts offline and optimistic exploration with the environment online. To estimate the dynamic model uncertainty, the authors utilize the model disagreement method which learns an ensemble of dynamic models.  This estimate uncertainty are utilized in two folds, it can serve as a penalty term during rollouts and as an incentive when interating with the environment online. The authors conduct experiments in environments like MuJoCo and DeepMind Control and demonstrate the proposed method achieves better sample efficiency and performance than other model-based RL baselines.

**Strengths:**

This paper is easy to follow and the proposed approach is also simple and flexible to use on existing model based RL methods. The authors have done extensive experiments and ablations to show their strengths in sample efficiency and model performance. I think the part of the success of this paper attributes to the good uncertainty estimation.

**Weaknesses:**

The good uncertainty estimation comes with the cost of extra computation, in Appendix D.6, there is an above 20\% increase for MBPO variant and around 40\% increase for the DreamerV3 variant. The test time comparison should also be discussed.

**Questions:**

Where do the authors see the main factor for the extra computation cost?  Is it the uncertainty calculation for all action candidates through the ensemble of dynamic models?

What options is available for trading off computation cost for uncertainty estimation performance?

In Figure 7, while the proposed method reached lower loss, this does not imply that the method achieved better exploration, which the authors claim in related work section, how can this be demonstrated?

---

> ### Author Response · Authors · 2023-11-17
> **Response to Reviewer EHTZ**
>
> We thank Reviewer EHTZ for the insightful feedback. We are encouraged that Reviewer EHTZ finds our method flexible, our experiments fruitful and results strong. We now address all of Reviewer EHTZ's concerns and questions below:
>
> > Weakness 1: The good uncertainty estimation comes with the cost of extra computation, in Appendix D.6, there is an above 20% increase for MBPO variant and around 40% increase for the DreamerV3 variant. The test time comparison should also be discussed.
>
> At test time, COPlanner, MBPO, and DreamerV3 are indistinguishable; all use the trained policy to directly interact with the environment, with the input being the observation and the output being the action. The UP-MPC is only used during the policy training phase, that is, interacting with the real environment to obtain real samples and using model rollouts to generate more imaginary samples. Therefore, the increased computational cost only exists during the training phase. During the test time, COPlanner does not incur any extra computational cost.
>
> > Question 1: Where do the authors see the main factor for the extra computation cost? Is it the uncertainty calculation for all action candidates through the ensemble of dynamic models?
>
> The extra computation cost mainly comes from two parts:
> 1. To estimate the model uncertainty of each model-generated sample, we train a dynamics model ensemble.
> 2. During the UP-MPC process, we need to perform 5-step planning for each action candidate, which also brings additional time consumption.
>
> > Question 2: What options is available for trading off computation cost for uncertainty estimation performance?
>
> Using alternative methods for estimating uncertainty in place of a model ensemble can improve computational efficiency. In Figure 12 of Appendix D.5, we compare methods that use different intrinsic rewards to guide exploration. The experiments show that although these methods can speed up training, their performance is not as good as the COPlanner using a model ensemble. Additionally, a potentially effective approach is to use a Bayesian dynamics model instead of a model ensemble for estimating model uncertainty. We plan to further explore and experiment with this approach in future work.
>
> > Question 3: In Figure 7, while the proposed method reached lower loss, this does not imply that the method achieved better exploration, which the authors claim in related work section, how can this be demonstrated?
>
> As mentioned in point five of our General Response, we add the evaluation of the sample diversity in the real sample buffer obtained by COPlanner-DreamerV3 and DreamerV3 in **Appendix D.7**. The experimental results shown in Figure 13 demonstrate that the sample diversity in the real sample buffer obtained by COPlanner-DreamerV3 is significantly better than that of DreamerV3, proving that COPlanner-DreamerV3 achieves better exploration of the real environment. For specific experimental details, please refer to Appendix D.7 in the updated paper.
>
>
> -----
> Thank you again for your effort in reviewing our paper! We are happy to answer any further questions.

---

> > ### Comment · Reviewer_EHTZ · 2023-11-20
> >
> > Thank you for addressing my questions!

---

### Author Response · Authors · 2023-11-17
**General response**

We thank all reviewers for their insightful questions and valuable feedback. We are encouraged that reviewers consider our paper  well-written and easy to follow (EHTZ, NmVv), the idea and the tackled problem interesting and important (NmVv, D4cs), and experiment results fruitful and strong (EHTZ, NmVv, D4cs). We have addressed all individual questions of reviewers in separate responses. As suggested by the reviewer, we have added more experiments in the updated paper. Below is a description of these experiments.

1. For proprioceptive continuous control tasks, we add comparisons with three more baselines including **M2AC, D4PG, and DreamerV3**. In **Figure 4**, following the suggestions of reviewers NmVv and D4cs, we include M2AC and D4PG as baselines. M2AC is a model-based method that avoids model prediction error misleading policy learning by filtering out samples with high model uncertainty after rollouts, and D4PG is a popular model-free method. Compared to M2AC and D4PG, our method also shows a significant advantage. Besides, in **Appendix D.1, Figure 9**, following reviewer D4cs's suggestion, we compare the performance of COPlanner-DreamerV3 and DreamerV3 on proprioceptive continuous control tasks. We conduct experiments in six medium-difficulty tasks in DMC. Compared to DreamerV3, COPlanner-DreamerV3 **improves performance by 40.7%**, and **sample efficiency is increased by about three times** in the best environment. This further demonstrates the strong performance of our method.
2. For visual control tasks, following reviewer D4cs's suggestion, we include a comparison with **DrQv2** in **Figure 5**. DrQv2 is a model-free visual RL method with very strong performance, serving as a common baseline in visual RL. Compared to DrQv2, COPlanner-DreamerV3 exhibits **a dominant advantage in both performance and sample efficiency**.
3. To present a clearer trend of how hyperparameter settings affect COPlanner's performance, we add two additional hyperparameter studies in **Appendix D.4, Figure 11**, following reviewer D4cs's suggestion. These studies are conducted on the proprioceptive control Cheetah-run task using COPlanner-MBPO and on the proprioceptive control Hopper-hop task using COPlanner-DreamerV3. The trends we observed are largely consistent with our previous experimental results.
4. In **Appendix D.7**, we evaluate the diversity of samples in the real sample buffer of COPlanner-DreamerV3 and DreamerV3 across four different visual control tasks to demonstrate that our method achieves better exploration. Following the work of Seo et al. [1], we evaluate the diversity of image samples by utilizing a k-nearest neighbor entropy estimator in the low-dimensional representation space of a randomly initialized encoder. The experimental results are shown in **Figure 13**. The results indicate that the real sample buffer collected by COPlanner-DreamerV3 has a higher k-nearest neighbor state entropy compared to DreamerV3, demonstrating that the samples in COPlanner-DreamerV3's real sample buffer are more diverse. This further suggests that COPlanner-DreamerV3 achieves better exploration in the real environment.
5. To further demonstrate our experimental results, we visualize the trajectories of policies learned by COPlanner-DreamerV3 and Dreamer V3 in the test environment after convergence of policy learning in **Appendix D.8**. From Figures 14 to 17, it can be seen that COPlanner-DreamerV3 learned more rational and stable behaviors compared to DreamerV3.



Reference:

[1]. "State Entropy Maximization with Random Encoders for Efficient Exploration." Seo et. al. ICML 2021.

---

### Meta-Review · Area_Chair_1FrY · 2023-12-13

**Metareview:**

This paper present a model-based reinforcement learning algorithm that combines uncertainty, optimism, and pessimism in a novel way to be able to compensate for the limitations of existing MBRL literature.

Overall the paper is interesting, novel and well written.

The authors addressed many points of the reviewers' feedback, and I believe that the current manuscript would be a worth contribution to the literature.

Out of curiosity: in the introduction, you make the statement that "learning an accurate dynamics model is critical". However, the [Lambert et al. 2020] which you cite seems to show otherwise. Thoughts?

**Justification For Why Not Higher Score:**

After the rebuttal, the reviewers are not fully satisfied with the answer of the authors. I agree with the reviewer's (minor) concerns.

**Justification For Why Not Lower Score:**

All the reviewers agree that the paper is worth of publications

---

### Decision · Program_Chairs · 2024-01-16

Accept (poster)